# High Expression Levels of miR-21-5p in Younger Hospitalized COVID-19 Patients Are Associated with Mortality and Critical Disease

**DOI:** 10.3390/ijms241210112

**Published:** 2023-06-14

**Authors:** Brandon Bautista-Becerril, Karol J. Nava-Quiroz, Evangelina Muñoz-Soria, Ángel Camarena, Ingrid Fricke-Galindo, Ivette Buendia-Roldan, Gloria Pérez-Rubio, Leslie Chavez-Galán, Karina Pérez-Torres, Fernanda Téllez-Quijada, Eduardo Márquez-García, Angelica Moncada-Morales, Rafael de Jesús Hernández-Zenteno, María Esther Jaime-Capetillo, Ramcés Falfán-Valencia

**Affiliations:** 1HLA Laboratory, Instituto Nacional de Enfermedades Respiratorias Ismael Cosio Villegas, Mexico City 14080, Mexico; brandon.bautistab@gmail.com (B.B.-B.); glofos@yahoo.com.mx (G.P.-R.); 2Sección de Estudios de Posgrado e Investigación, Escuela Superior de Medicina, Instituto Politécnico Nacional, Mexico City 11340, Mexico; 3Translational Research Laboratory on Aging and Pulmonary Fibrosis, Instituto Nacional de Enfermedades Respiratorias Ismael Cosio Villegas, Mexico City 14080, Mexico; 4Laboratory of Integrative Immunology, Instituto Nacional de Enfermedades Respiratorias Ismael Cosio Villegas, Mexico City 14080, Mexico; 5Unidad de Biología Molecular, Instituto Nacional de Enfermedades Respiratorias Ismael Cosio Villegas, Mexico City 14080, Mexico; 6COPD Clinic, Instituto Nacional de Enfermedades Respiratorias Ismael Cosio Villegas, Mexico City 14080, Mexico; 7Clinical Laboratory Service, Instituto Nacional de Enfermedades Respiratorias Ismael Cosio Villegas, Mexico City 14080, Mexico; tetejaime_2001@yahoo.com

**Keywords:** microRNAs, COVID-19, viral infections, mortality, mechanical ventilation, miR-21-5p

## Abstract

In COVID-19, critical disease and invasive mechanical ventilation (IMV) increase the risk of death, mainly in patients over 60 years of age. Objectives: To find the relationship between miR-21-5p and miR-146a-5p in terms of the severity, IMV, and mortality in hospitalized COVID-19 patients younger than 55 years of age. Methods: The patients were stratified according to disease severity using the IDSA/WHO criteria for severe and critical COVID-19 and subclassified into critical non-survivors and critical survivors. Results: Ninety-seven severe/critical COVID-19 patients were included; 81.3% of the deceased were male and 18.8% were female. Higher expression miR-21-5p levels were associated as follows: severe vs. critical disease (*p* = 0.007, FC = 0.498), PaO_2_/FiO_2_ index, mild vs. severe (*p* = 0.027, FC = 0.558), and survivors vs. non-survivors (*p* = 0.03, FC = 0.463). Moreover, we identified correlations with clinical variables: CRP (rho = −0.54, *p* < 0.001), D-dimer (rho = −0.47, *p* < 0.05), related to damage in the kidney (rho = 0.60, *p* < 0.001), liver (rho = 0.41, *p* < 0.05), and lung (rho = 0.54, *p* < 0.001). Finally, miR-21-5p thresholds were calculated according to severity (8.191), IMV (8.191), and mortality (8.237); these values increased the risk of developing a critical disease (OR = 4.19), the need for IMV (OR = 5.63), and death (OR = 6.00). Conclusion: Increased expression levels of miR-21-5p are related to worse outcome of COVID-19 in younger hospitalized patients.

## 1. Introduction

The COVID-19 pandemic represents one of the most significant challenges in the history of public health. As of 5 October 2022, there were more than 620 million cases worldwide and more than 6.55 million deaths [1]. Approximately 20% of patients develop a severe disease requiring hospital admission, and more than 5% require intensive care unit (ICU) attention, with a variable ICU mortality rate ranging from 25 to 50% [2]. The main risk factors associated with more severe disease and death are patients older than 60 years [3], body mass index (BMI) > 30 kg/cm^2^ [4], type 2 diabetes mellitus (T2DM) [5], and systemic arterial hypertension (SAH) [6].

Between June–September 2021, some countries from Latin America and Europe showed a critical increase in infections, and a change in the demographics of patients who developed severe illness and/or death was evidenced; the median age decreased from 60 to 40 years [7,8,9,10]. Furthermore, most patients did not show the same risk factors as those older than 60 [7].

In this context, many studies have delved into less-studied risk factors associated with a high risk of developing severe forms of the disease and death. In addition to classical risk factors, genetic variants in several loci, such as *TNFRSF1A* [11], *IFNAR2* [12], and *SERPINE1* [13], have been described by our group. However, studies evaluating the participation of miRNAs in the severity and mortality of COVID-19 are scanty.

miRNAs are a relatively recently described group of small RNA molecules of 20 to 24 nucleotides that do not code for proteins but regulate more than 60% of all genes [14,15]. They can be part of complex regulatory networks in the gene expression of pathophysiological processes in progressing to severe forms of COVID-19 [16,17].

Therefore, we aimed to determine if alterations in miR-21-5p and miR-146a-5p levels are associated with severity and mortality in subjects under 55 years of age who were hospitalized for COVID-19. We selected these miRNAs due to a large number of target genes and pathways related to inflammation, heart, lung, and kidney damage, coagulopathy, pulmonary fibrosis, and sepsis, leading causes of severe illness, the need for mechanical ventilation, and death in COVID-19 patients, which we addressed in an extensive previous review [18]; the miRNA characteristics are addressed in the Appendix A, Appendix A: Characteristics on miR-25-5p and miR-146a-5p.

## 2. Results

The study population consisted of 128 peripheral blood samples from Mexican mestizo COVID-19 patients, and 31 patients older than 55 years were excluded, thus keeping 97 eligible samples. The selected samples were divided into two groups following the principle of proportionality according to the severity variable of the hospital population, leaving a group of 65 critical patients and 32 severe patients (Appendix A, Appendix A: Stratification of the study population).

The description of the primary demographic data according to the severity of the disease, clinical characteristics, comorbidities, and pre-existing conditions is shown in Table 1. A total of 97 samples were obtained: 73 male and 24 female. For every woman in critical condition, there were 3.08 men, with a median age of 43 years.

On the other hand, when stratifying into survivors and non-survivors, it was found that 81.3% of the deceased patients were male and 18.8% were female, with a median age of 48.5 years (Table 2). Additionally, a generalized regression model was made, taking age and BMI as possible confounding variables; however, no significant results were found (*p* = 0.788, *p* = 0.769, respectively), so age or BMI did not modify the expression levels of miRNAs.

### 2.1. The miR-21-5p and miR-146a-5p and Their Relation with Mortality in Patients with COVID-19

A significant difference in miR-21-5p expression was found in surviving patients compared with non-surviving patients (*p* = 0.03), with a difference in the expression between these two groups (or fold change, FC) of 0.463 in favor of the group of non-surviving patients (Figure 1A). For miR-146a-5p, the non-survivors group (FC = 0.237) was increased, but no significant difference was found (*p* = 0.21, Figure 1B).

Owing to the difference in the age observed among surviving patients (Table 2), we performed a Cox regression analysis, including age and expression levels of miR-21-5p as co-variables; however, no significance was found (Appendix A, Appendix A: Cox regression analysis for the multivariable assessment of mortality). Thus, these are independent variables affecting the mortality, which agrees with the lack of influence of age in the miRNA expression.

### 2.2. The miR-21-5p and miR-146a-5p Expression between Patients with Severe and Critical COVID-19

miR-21-5p had a statistically significant difference between these groups (*p* = 0.007) and an FC = 0.498 in favor of patients with a critical clinical disease (Figure 2A). The miR-146a-5p showed an increase in expression (FC = 0.161) and no statistically significant difference between the two study groups (*p* = 0.18, Figure 2B).

Subsequently, we stratified patients into three groups, severe, critical survivors (CS), and critical non-survivors (CNS). miR-21-5p presented significant differences between the three groups studied (*p* = 0.016). Subsequently, Dunn’s test revealed a significant difference in the expression levels between the severe and CNS groups (*p* = 0.005). However, no differences were found in the CS, CNS, or severe patients (*p* = 0.061, *p* = 0.329, respectively). (Figure 3A). The miR-146-5p did not express significant differences between the three groups studied (*p* = 0.320) nor Dunn’s subsequent test (Figure 3B).

### 2.3. The miR-21-5p and miR-146a-5p and Their Relation to a Grade of Acute Respiratory Distress Syndrome in COVID-19

The miR-21-5p presented significant differences between the three groups studied (*p* = 0.027). Subsequently, in Dunn’s test, significant differences were found between the mild–severe and moderate–severe groups (*p* = 0.014 and *p* = 0.018, respectively), and for the mild–moderate groups, no significant differences were found (*p* = 0.748) (Figure 4A). Regarding miR-146-5p, no significant differences were found between the three groups studied (*p* = 0.460) or between pairs when performing Dunn’s tests (Figure 4B).

### 2.4. miRNA Relation to Comorbidities in Patients with Severe and Critical Disease in COVID-19

We compared the difference in the expression of miR-21-5p and miR-146a-5p between the main comorbidities of the patients regardless of the severity of COVID-19. The expression values of both miRNAs were raised in patients with comorbidities (T2DM, SAH, and tobacco smoking) compared with those without comorbidities; however, neither showed a significant difference between the groups; therefore, the relative expression of these miRNAs could not be affected by either comorbidity (Appendix A, Appendix A: miR-21-5p and miR-146a-5p and their relation with comorbidities in patients with severe and critical disease in COVID-19).

### 2.5. Correlation between miRNA Relative Expression and Clinical and Laboratory Variables

Correlations were sought using more than 32 clinical and laboratory datasets. The miR-21-5p showed significant correlations with variables related to coagulation, lung damage, kidney impairment, lipids, and glucose metabolism. In contrast, miR-146a only showed significant correlations with variables related to coagulation pathways. A summary of the statistically significant correlations can be found in Figure 5. A complete list of included variables can be found in the Appendix A, Appendix A: Clinical and laboratory variables included in the correlation tests.

In the ninety-seven subjects, the most representative correlations with the miR-21-5p were with D-dimer (DD), ferritin (FTL), total cholesterol (TC), sodium (Na), glucose, and blood urea nitrogen (BUN). However, they did not show a strong correlation (rho = 0.24–0.38) but displayed statistical significance (Figure 5). For miR-146a-5p, only platelets and prothrombin were correlated (*p* = 0.026 and 0.048, respectively; Appendix A, Appendix A: Statistically significant correlations between miR-146a-5p and clinical variables). Among clinical variables, the notable correlations and statistically significant (*p* < 0.01) were urea/BUN (rho = 0.99), total proteins/albumin (rho = 0.68), DD/total cholesterol (rho = −0.59), and PaO_2_/FiO_2_ ratio/days of IMV (rho = 0.48); these and other correlations are summarized in Figure 5A.

In addition, we found that in the non-survivor patients, the miR-21-5p correlations with variables such as glucose, urea, and BUN were stronger than in the general group (rho = 0.43, 0.50, and 0.48, respectively), and C reactive protein (CRP) and procalcitonin (PCT) showed a statistically and strong correlation in this group (rho = −0.54 *p* < 0.01, rho = 0.54 *p* < 0.01, respectively). These and other correlations are summarized in Figure 5B.

### 2.6. Receiver Operating Characteristic Curve Analysis of miR-21-5p and miR-146a-5p and Mortality in COVID-19 Patients

ROC curve analyses showed a high ability to distinguish non-surviving patients based on miR-21-5p expression levels (cut-off = 8.237, AUC = 63.3%, 95% CI = 0.512–0.761, *p* = 0.03) (Figure 6A). However, miR-146a-5p and mortality showed no statistical significance to distinguish non-survivors (cut-off = 5.884, AUC = 57.9%, 95% CI = 0.462–0.671, *p* = 0.206) (Figure 6B).

Similarly, miR-21-5p showed a high ability to distinguish patients with more severe disease (cut-off = 8.191, AUC:66.9%, 95% CI: 0.562–0.777, *p* = 0.007) (Figure 6C) and the need for IMV (cut-off = 8.191, AUC:67.5%, 95% CI: 0.568–0.781, *p* = 0.005) (Figure 6E). In contrast, miR-146a-5p did not show significant results in distinguishing patients with more severe disease or the need for IMV (cut-off = 9.148, AUC: 58.5%, 95% CI: 0.464–0.705, *p* = 0.177, cut-off = 4.851, AUC: 56.2%, 95% CI: 0.440–0.685, *p* = 0.315, respectively) (Figure 6D,F). In addition, the best cut-off of the expression levels for miR-21-5p and miR-146a-5p for mortality, disease severity, and the need for IMV were calculated following the coordinates of the ROC curves and using the tallest index Youden.

With the previously selected cut-off, miR-21-5p was shown to increase mortality risk by 4.1-fold, the risk of developing critical illness by 5.6-fold, and the need for IMV by 6.0-fold. However, miR-146a-5p did not show significant results (Table 3).

## 3. Discussion

Our study aimed to identify the expression and relationship between miR-21-5p and miR-146a-5p, which have been linked to multiple inflammatory pathways and antiviral immune responses in patients with varying degrees of COVID-19 severity [19,20,21,22].

We found that miR-21-5p levels are associated with mortality, severity, and PaO_2_/FiO_2_ index, as well as some correlations with clinical variables in the severity and mortality risk of COVID-19. Moreover, miR-21-5p could distinguish patients with a higher risk of death, severe disease, and patients needing an IMV.

In the current study, the patients with severe or critical disease were younger than 55 years of age, and the critical disease patients were older than the severe group; moreover, most of the critical patients were overweight or obese, agreeing with previous reports [23,24,25]. Interestingly, comorbidities such as T2DM and SAH were more frequent in severe patients than in critical. In addition, all the non-survivor patients required IMV and used it for a longer time.

In this study, we show that the relative expression levels of miR-146a-5p (although not statistically significant) are elevated in COVID-19 patients with a worse clinical outcome. In a smaller population, Tang and collaborators found that miR-146a-5p expression was significantly decreased in severe patients [26].

For miR-21-5p, we found that higher expression levels were associated with mortality, worst critical disease, and a lower PaO_2_/FiO_2_ index. Garg et al. mention that miR-21-5p expression is increased in the COVID-19 group compared with healthy controls and patients with influenza ARDS, suggesting that the upregulation of miR-21 in COVID-19 survivors could be a predictor of inflammation and chronic myocardial damage [22].

Our results could be explained by the significant miR-21-5p relation with the pro-inflammatory effect and its ability to promote the activation of the NLR family pyrin domain containing 3 (NLRP3) NLRP3 inflammasome, thus enabling the secretion of IL-1β and the cleavage of caspase-1, as well as the activation of gasdermin D that potentiates the activation of inflammasomes and the production of IL-6 and TNF-α [27,28,29]. Moreover, the miR-21 controls mRNAs such as the programmed cell death receptor 4 (PDCD4), E-cadherin, tissue inhibitor of metalloproteinase 2, which is responsible for regulating IL-1B, IL-18, IL-6, IL-33, NF-κB, TNF-α, and STAT 3/5, and Caspase-1/4/5/11 [29,30,31]. Together, these molecules are intensely involved in lung inflammation, repair, regeneration, and remodeling of the injured respiratory tissue, including stressed or injured epithelium, increased smooth muscle mass, subepithelial fibrosis, enlargement of goblet cells and submucosal glands, hyperemia with increased vascularization of subepithelial tissues, thickening of the basement membrane, and extracellular matrix deposition [28,32,33,34,35]. These data support our results that upregulated miR-21-5p expression levels lead to the most critical disease evidenced by a lower PaO_2_/FiO_2_ index, more days of IMV, and death.

Almost all risk factors could result in different expressions of specific miRNAs in peripheral blood, including those older than 60 years, obesity, SAH, and T2DM [36,37]. Our results show that these comorbidities did not modify the miR-21-5p expression levels. On the other hand, although the sex of the patients could significantly influence the expression of these miRNAs, we did not find significant differences in the comparisons made.

Clinical variables such as D-dimer [38], procalcitonin [39], glucose [40], creatinine [41], and urea [42] have demonstrated an association with coagulation pathways and lung, kidney, and liver damage, which could lead to more severe disease and death in COVID-19 patients. Upregulated miR-21-5p expression levels were statistically correlated with procalcitonin, creatinine, urea, and thromboplastin-D-dimer index in non-survivor patients; albumin, D-dimer, and ferritin levels were correlated in the whole population. On the other hand, Giannella et al. described that the upregulated miR-21-5p correlated with variables such as IL-6 and CPK, which were involved in an inflammatory process. Of note, the CRP levels show an inverse correlation with the miR-21-5p compared with other publications [43]. In this sense, we found an association with clinical variables related to inflammation, coagulation, kidney, hepatic, and pulmonary damage pathways.

Different miRNA cut-offs in COVID-19 can predict hospitalized patients’ outcomes [20]. We found that miR-21-5p expression levels around 8.2 could distinguish more severe disease, the requirement for IMV, and death among COVID-19 patients. It should be noted that these similar cut-offs between our three main variables further demonstrate that miR-21-5p expression is maintained during the progression to more severe disease and provide the ability to predict which patients could develop a more critical illness. Even though miR-21 has been evaluated as a specific predictive or prognostic biomarker in many diseases [44], the results are inconclusive. Still, many other investigations have suggested a close relation with lung involvement in the COVID-19 pathophysiology [45].

In a recent clinical study, our group identified that the age >65 years has an increased death risk (OR = 2.7) [46]. In the current study, the miR-21-5p cut-off >8 in patients under 55 showed a higher risk of developing a worse condition requiring IMV. Interestingly, the odds ratios are lower in the mortality (OR = 4.1229) compared with the severity of the disease (OR = 5.639) and the need for IMV (OR = 6.007); therefore, we suggest that early intervention could modify (decrease) the risk of death.

We found that miR-146a-5p had a statistically positive correlation with platelet count in patients with COVID-19; mir-146a-5p has previously been described to have interactions with molecules such as platelet-derived growth factor receptor alpha (PDGFRA), which increases the transduction of platelet signals and stimulates adhesion between platelets and the endothelium. In addition, it has been linked to patients with severe COVID-19 and thromboembolic events, as well as in patients who required admission to the intensive care unit (ICU) [47,48].

On the other hand, miR-21-5p had significant correlations with markers related to kidney damage, and it has been previously described that this miRNA plays an important role in regulating pathways related to programmed cell death protein 4 (PDCD4) that inhibit the translation of initiation factor eIF4A, a helicase responsible for regulating the cell cycle and apoptosis. This promotes NF-kB activation and suppresses IL-10 translation and the activation of caspase 3 [49,50], which together are related to atrophy, dilation, and swelling of the renal tubules, accompanied by expressed tubular necrosis, initially expressed as elevation in the increase in BUN and creatinine levels in serum [51]. Similar results were reported by Chen et al., where higher urinary levels of miR-21 were related to creatinine elevations, tubulointerstitial injury, and renal fibrosis at eight days in mice models with renal-induced damage [52]. On the other hand, miR-21 overexpression was related to increased plasma creatinine levels, progressively increasing protein and albumin excretion rates, with low serum levels, proximal tubular damage including pronounced dilation, the formation of apical bullae, and the loss of the brush border of renal cells [53].

Although there is little information on the relationships of miR-21-5p with other laboratory variables, miR-21 has previously been shown to bind to the 5′-UTR of IκB Kinase (IKKB) protein and TNF-α by the interaction between the maternally expressed protein 3 (MEG3) and miR-21. IKKB phosphorylates the inhibitor of NF-κB (IκBα) and promotes its degradation, thus activating NF-κB p65 and promoting its translocation into the nucleus, which acts as a transcription factor to activate the transcription of inflammatory-related genes, such as IL-1 and IL-6, and TNF-α, thus promoting mitochondrial activity and NADPH oxidase expression, which are the primary source of endogenous ROS [54]. In addition to elevating acute phase proteins such as C-reactive protein [55] and fibroblast growth factor 18 (FGF18), an increased concentration of fibroblast growth factor, granulocyte colony-stimulating factor (G-CSF), granulocytes, and monocytes colony-stimulating factor (GM-CSF) has been linked to hospitalized COVID-19 patients compared with healthy controls [56]. Moreover, COVID-19 patients who died exhibited significantly elevated levels of IFN-α, IFN-λ, and IL-1Ra, as well as chemokines associated with monocytes and T cell recruitment and survival such as CCL1, CLL2, macrophage colony-stimulating factor (M-CSF), IL-2, IL-16, and CCL21, within the first 12 days from symptom onset [57].

Our study is not exempt from limitations; firstly, it was elaborated in a third-level hospital that only received patients with severe or critical COVID-19 disease and not patients with mild or asymptomatic disease. Moreover, while other studies evaluated a large number of miRNAs, we studied two miRNAs that, in a previous review, showed significant differences in COVID-19 patients with extensive associations with molecules that support the development of severe forms of COVID-19 [18]. On the other hand, we did not collect data from current healthy controls in this study since we had to demonstrate that the controls had had a positive test for SARS-CoV-2 to compare under similar circumstances (differences in viral variants mainly). Moreover, laboratory testing was limited to people with severe or critical illnesses at recruitment time. Furthermore, our study focuses on a single biological sample obtained at the disease onset so that the treatment did not change the interpretation of the results. We will consider the above limitations for future studies and in the development of a new longitudinal study design.

As far as we know, this is the first study with a large number of recruited patients that seeks to recognize the relations between the studied miRNAs in the plasma of patients with COVID-19 not older than 55 years. Moreover, we included an important number of critical patients, identifying associations with disease severity, the need for IMV, and mortality. Moreover, we established cut-offs that could distinguish patients with more risk of developing severe disease and calculated the odds ratios that show the critical role of the studied miRNAs in COVID-19.

## 4. Materials and Methods

### 4.1. Study Population

This study included Mexican mestizo men and women between 22 to 55 years admitted to the hospital with a diagnosis of SARS-CoV-2 via a nasopharyngeal swab PCR test at the Instituto Nacional de Enfermedades Respiratorias Ismael Cosio Villegas (INER) during the third wave of COVID-19 in Mexico (July–August 2021). All patients were evaluated by specialist pulmonologists using clinical and laboratory criteria to stratify them according to the severity of the disease using the Infection Disease Society of America (IDSA) [58] complemented with the WHO criteria [59] in the following groups:

Severe: Clinical signs of pneumonia (fever, cough, dyspnea, and tachypnea) plus any of the following: respiratory rate > 30 inspirations/min, severe respiratory distress, or SpO_2_ < 90% on room air, but no requirement for IMV.

Critical: The patient presented acute respiratory distress syndrome (ARDS) and required immediate IMV. Moreover, the patient presented signs of sepsis, septic shock, or life-threatening multi-organ damage.

The ARDS is classified as follows.
Mild ARDS level: 200 mmHg < PaO_2_/FiO_2_ ≤ 300 mmHg.Moderate ARDS: 100 mmHg < PaO_2_/FiO_2_ ≤ 200 mmHg.Severe ARDS: PaO_2_/FiO_2_ ≤ 100 mmHg.

Subsequently, the patients received follow-up assessments during their hospital stay to report the severity of the disease until their death or recovery and discharge from the hospital. Complete demographic, clinical, and laboratory data were manually extracted from electronic medical records. As exclusion criteria, patients who decided to discharge themselves voluntarily, whose biological sample did not contain the necessary quantity and quality, patients with coinfection of another virus proven by PCR tests, and patients diagnosed with cancer or undergoing treatment were excluded to avoid potential conflicts when reading the results of the miRNAs studied, which have been previously studied in cancer and other diseases.

### 4.2. Statement of Ethics

The study was conducted in full compliance with the Declaration of Helsinki. The institutional ethics committee approved the study protocol (approbation number: C53-20). The participants, or their legal representatives, gave their written informed consent.

### 4.3. Obtaining Blood Samples and RNA Isolation

On the patient’s arrival at the hospital for admission, blood samples were collected in ethylenediaminetetraacetic acid (EDTA) tubes. Subsequently, the plasma was separated, and the presence of hemolysis in the plasma was evaluated through a first visual inspection and a second review based on spectrophotometry with the NanoDrop™ 2000/2000c instrument (Id: ND-2000, Thermo Scientific™, Foster City, CA, USA) with readings at wavelengths from 350 to 650 nm. Samples are considered hemolyzed if the reading at 414 nm exceeds a value of 0.2 [60,61]. Samples that did not meet this requirement were discarded for this study. Afterward, RNA extraction was performed with the miRNeasy serum/plasma Advanced Kit (Qiagen, Hilden, Germany). For normalization, synthetic Caenorhabditis elegans 39-3p (cel-miR-39-3p) was added as an external reference miRNA (1.6 × 10^8^ copies/μL). The next step was quantifying the RNA obtained via spectrophotometry at 260 nm wavelength using the NanoDrop™ 2000/2000c instrument (Id: ND-2000, Thermo Scientific™, Foster City, CA, USA). The contamination with organic compounds and proteins was established using the ratio of the readings 260/230 and 260/280, respectively; the samples were considered free of contaminants when the relationship was between 1.7–2.0. Then, they were stored at −20 °C until use.

### 4.4. RT-PCR and Polymerase Chain Reaction for miRNA Analysis

The cDNA synthesis was performed using a TaqMan^®^ Advanced miRNA cDNA Synthesis Kit (Id: A28007, Applied Biosystems™ Foster City, CA, USA) in 10 μL following the manufacturer’s specifications. Afterward, miRNAs were amplified with the following TaqMan probes: hsa-miR-21-5p (Id: A25576 477975_mir, Applied Biosystems™ Foster City, CA, USA) and hsa-miR-146a-5p (A25576 478399_mir, Applied Biosystems™ Foster City, CA, USA) using the Cel-miR-39-3p (Id: 4427975 000200, Applied Biosystems™, Foster City, CA, USA) probe as an endogenous gene. The reaction mixture was carried out in a final volume of 25 µL in a StepOne™ Real-Time PCR System (Id: 4376357, Thermo Scientific™, Foster City, CA, USA) following the manufacturer’s instructions. In addition, the expression value of Cel-miR-39 was obtained as the normalization factor. The relative quantification of each miRNA was calculated using the formula 2^–∆Ct^, and the value of change or fold change was calculated using the formula 2^–∆∆Ct^.

### 4.5. Statistical Analysis

All statistical analyses were performed with SPSS 21.0 software and RStudio 4.2.0 (libraries: ggplot2, corrplot, pROC, Hmisc, tidyverse, rstatix, and stringi). The normality of the data distribution was calculated using the Kolmogorov–Smirnov test; for the quantitative variables, the median and interquartile ranges were used. Then, to compare the expression levels of the miRNAs between the groups of the two categories, a Mann–Whitney–Wilcoxon U test was conducted; and the Kruskal–Wallis test was conducted in three categories. A Cox regression analysis was performed for the multivariable assessment of mortality. In addition, Dunn’s multiple comparisons test was carried out to compare pairs; both tests had a statistical significance of 0.05. Spearman’s rank correlation coefficient was calculated to assess the relationship between plasma miRNA levels with different clinical variables and laboratory values with a statistical significance of 0.05. For the discrimination of groups by selected miRNA levels, receptor operating characteristic (ROC) curves were used, and the area under the curve (AUC) was calculated. Subsequently, the chosen miRNA cut-off was calculated using Youden’s indices, and odds ratios (OR) and 95% confidence intervals (CI) were estimated.

## 5. Conclusions

The increased expression levels of miR-21-5p are related to worse outcome of COVID-19 in younger hospitalized Mexican mestizo patients.

## Figures and Tables

**Figure 1 ijms-24-10112-f001:**
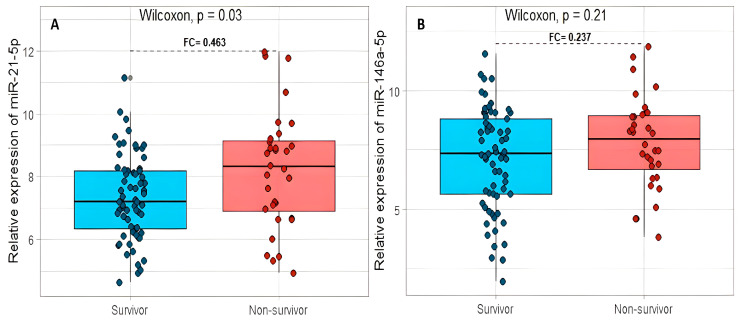
The relation between the relative expression levels of miR-21-5p (**A**) and miR-146a-5p (**B**) with the mortality in COVID-19 patients. FC: fold change.

**Figure 2 ijms-24-10112-f002:**
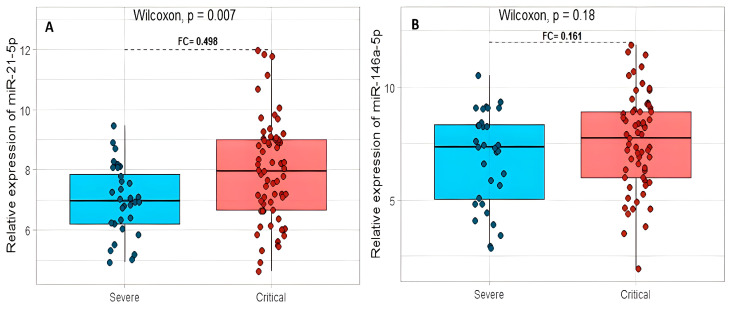
The relation between the relative expression levels of miR-21-5p (**A**) and miR-146a-5p (**B**) with severe and critical COVID-19 patients. FC: fold change.

**Figure 3 ijms-24-10112-f003:**
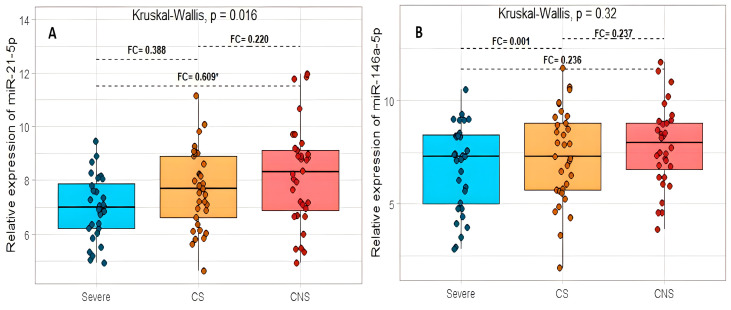
The relation between the relative expression levels of miR-21-5p (**A**) and miR-146a-5p (**B**) with the severe critical survivor and critical non-survivor COVID-19 patients. Critical survivor (CS); critical non-survivor (CNS). FC: fold change. * *p* < 0.05.

**Figure 4 ijms-24-10112-f004:**
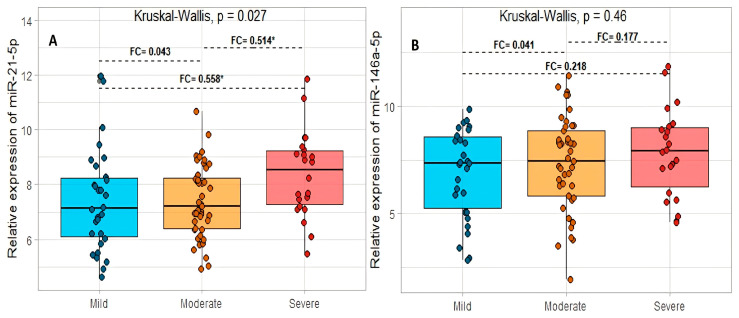
The relation between the relative expression levels of miR-21-5p (**A**) and miR-146a-5p (**B**) and the grade of acute respiratory distress syndrome in COVID-19 patients. FC: fold change. * *p* < 0.05.

**Figure 5 ijms-24-10112-f005:**
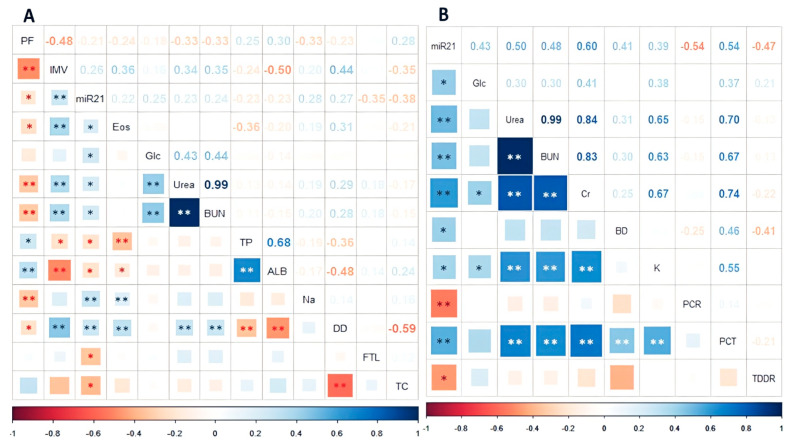
The miR-21-5p correlation with clinical and laboratory variables in all patients (**A**), and the miR-21-5p correlation in thirty-two non-survivor patients (**B**), with the PaO_2_/FiO_2_ ratio at hospital admission (PF), days of invasive mechanical ventilation (IMV), eosinophils count (Eos), glucose (Glc), urea, blood urea nitrogen (BUN), total proteins (TP), albumin (ALB), sodium (Na), D-dimer (DD), thromboplastin/D-dimer ratio (TDDR), ferritin (FTL), total cholesterol (TC), bilirubin direct (BD), protein C reactive (PCR), and procalcitonin (PCT) in COVID-19. * *p* < 0.05. ** *p* < 0.01.

**Figure 6 ijms-24-10112-f006:**
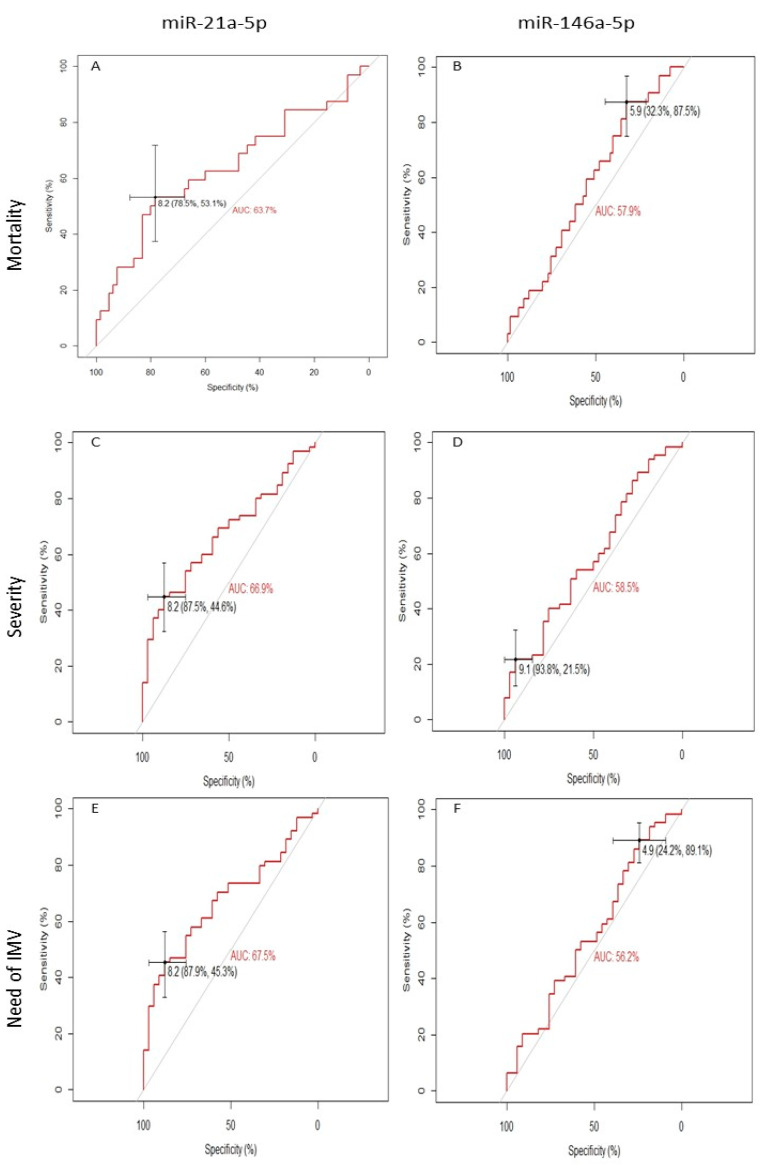
Receiver operating characteristic curve analysis and area under the curve of miR-21-5p and mortality (**A**); miR-21-5p and disease severity (**B**); and miR-21-5p and the need for IMV (**C**). In addition, miR-146a-5p and mortality (**D**); miR-146a-5p and disease severity (**E**); miR-146a-5p and the need for IMV (**F**) data are shown. Each variable’s cut-offs and sensitivity and specificity between parentheses are shown. AUC: area under the curve.

**Table 1 ijms-24-10112-t001:** Demographic and comorbid characteristics by severe and critical group type.

Characteristics	Severe *n* = 32	Critical *n* = 65	*p*-Value
Age, years	41.5 (31–46)	43 (38–52)	0.047
Sex (*n*, %)			
Female	11 (34.4)	13 (20)	0.139
Male	21 (65.6)	52 (80)	
BMI, kg/m^2^	27.91 (24.55–30.39)	29.40 (27.10–34.16)	0.027
BMI classification (*n*, %)			
Normal weight	12 (37.5)	13 (20)	0.084
Overweight	9 (28.1)	20 (30.8)	0.974
Obesity	11 (34.4)	32 (49.2)	0.196
Symptoms onset, days	8 (7–10)	8 (7–12)	0.266
PaO_2_/FiO_2_ ratio	210 (161–248)	129 (80–169)	0.001
Classification PaO_2_/FiO_2_ ratio (*n*, %)			
Mild	18 (56.2)	12 (18.5)	0.003
Moderate	14 (43.8)	31 (47.7)	0.829
Severe	0 (0)	22 (33.8)	0.001
Comorbidities			
T2DM	10 (31.3)	15 (23.1)	0.461
SAH	8 (25)	14 (21.5)	0.797
Cardiovascular disease	1 (3.1)	1 (1.5)	0.533
Respiratory disease	3 (9.4)	1 (1.5)	0.115
Tobacco smoking	11 (34.4)	22 (33.8)	0.566

Continuous data are presented as median (interquartile range, IQR) and categorical data as number and frequency in percentage (%). Statistical tests employed for the comparisons: Mann–Whitney U test and Fisher’s exact test. BMI, body mass index; Classification of BMI: Normal weight (BMI 18.5–24.9), overweight (BMI 25.0–29.9), and obesity (BMI > 30); SAH, systemic arterial hypertension; and T2DM, Type 2 diabetes mellitus. PaO_2_/FiO_2_ ratio at hospital admission, mmHg.

**Table 2 ijms-24-10112-t002:** Demographic and clinical characteristics of survivors and non-survivors.

Characteristics	Survivors *n* = 65	Non-Survivors *n* = 32	*p*-Value
Age, years	42 (32–48)	48.5 (41–52)	0.007
Sex (*n*, %)			
Female	18 (27.7)	6 (18.8)	0.454
Male	47 (72.3)	26 (81.2)	
PaO_2_/FiO_2_ ratio	160 (106–225)	143.65 (68.42–195)	0.143
Classification PaO_2_/FiO_2_ ratio (*n*, %)			
Mild	22 (33.8)	8 (25)	0.484
Moderate	31 (47.7)	14 (43.8)	0.829
Severe	12 (18.5)	10 (31.3)	0.198
IMV	32 (49.2)	32 (100)	0.001
Length IMV, days	6 (1–16)	23.50 (15–34)	0.001

Continuous data are presented as median (interquartile range, IQR) and categorical data as number and frequency in percentage (%). Statistical tests employed for the comparisons: Mann–Whitney U test and Fisher’s exact test. IMV, invasive mechanical ventilation. PaO_2_/FiO_2_ ratio at hospital admission, mmHg.

**Table 3 ijms-24-10112-t003:** Odds ratios of miR-21-5p and miR-146a-5p and the patient outcomes.

**miR-21-5p**				
**Variable**	**Cut-Off**	**Odds Ratio**	**95% CI**	** *p* ** **-Value**
Mortality	8.237	4.129	1.658–10.227	0.020
Severity	8.191	5.639	1.774–17.914	0.002
Need of IMV	8.191	6.007	1.892–19.075	0.001
**miR-146a-5p**				
**Variable**	**Cut-Off**	**Odds Ratio**	**95% CI**	** *p* ** **-Value**
Mortality	5.884	2.577	0.869–7.640	0.081
Severity	9.148	2.654	0.703–10.010	0.138
Need of IMV	4.851	2.480	0.755–6.696	0.140

IMV: invasive mechanical ventilation. Statistical tests were employed for the comparisons. Fisher’s exact test.

## Data Availability

The authors confirm that the raw data to support this study’s conclusions are included in the manuscript. The corresponding author will provide more information to any qualified researcher upon a reasonable request.

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
