# Peer review of "High Expression Levels of miR-21-5p in Younger Hospitalized COVID-19 Patients Are Associated with Mortality and Critical Disease"

_ijms, 2023, doi:10.3390/ijms241210112_

Round 1

Reviewer 1 Report

The authors have nicely demonstrated the correlation of high expression levels of miR-21-5p with the worst outcome of young hospitalized patients. 

There are some minor comments to improve the manuscript:

1. The figure legends are not written properly and should be improved so that the readers can understand even with just reading the figure legends. 

2. in figure 3, p values in the legend would help to better understand the figure. Also, atleast showing star for p value in the figures would be good as well. 

3. Line 159 should be rephrased. The english is not correct. 

4. Check for the abbreviations. Sometimes they are not explained when they are first mentioned in the manuscript. 

5. One general comment for the manuscript - the authors should also comment on the ethinicity of the covid patients who have been analysed. Without that, it is difficult to give a concrete conclusion for the analysis. 

Author Response

  1. The figure legends are not written properly and should be improved so that the readers can understand even with just reading the figure legends. 

A: We thank the reviewer for the observations; we have done the changes and hope that now are clearer.

  1. in figure 3, p values in the legend would help to better understand the figure. Also, atleast showing star for p value in the figures would be good as well. 

A: The statically significant p-values (according to Dunn´s test) are marked with a “*” next to the Fold Changes values, which should not be confused with the p-values shown at the top of the image. We apologize for the confusion.

  1. Line 159 should be rephrased. The English is not correct. 

A: We have made the change; we thank the reviewer.

  1. Check for the abbreviations. Sometimes they are not explained when they are first mentioned in the manuscript. 

A: We appreciate the reviewer’s comment, and that’s right, we had missed some abbreviations and now we have made the manuscript changes.

  1. One general comment for the manuscript - the authors should also comment on the ethinicity of the covid patients who have been analysed. Without that, it is difficult to give a concrete conclusion for the analysis. 

A: Again, we thank the comment, and we add this information in the Materials and Methods, Discussion, and Results sections.

Reviewer 2 Report

The study presented by Bautista-Becerril et al. tests the association of two very specific miRNAs (miR-21-5p and miR-146a-5p) with COVID-19 disease outcomes with an emphasis on severe disease manifestations and mortality. Overall, it is a well-designed study that led to a very well-written manuscript. The flow of the manuscript and the study's questions as well as the findings are cohesively presented and the data visualization is adequate. The cohort in which the questions were addressed is well-balanced and the conclusions (that miR-21-5p is causal to COVID-19 severity) are therefore sane. I think this study should be accepted after major revision, and here are my suggestions on what the authors should address for the revised manuscript: 

- The focus on those two aforementioned miRNAs is well explained from a theoretical perspective in the cited review [18] by the same group. However, I think it would be more appropriate if they have a data-driven explanation as to why they narrowed down to these two miRNAs compared to others. Therefore, I would like to see in the revised manuscript an expression analysis of all miRNAs expressed in the cohort and a correlation of each to different metrics of severity, as well as random metrics to validate the selection of those two miRNAs versus others. The survival conclusions should also be validated with the coxph statistical method. 

- The resolution of the figures should be improved (it looks like a copy-paste from R studio) 

I think the rest of the manuscript (both in writing and in the analysis) is good. I would be happy to have a second look at the revised manuscript, pending the editor's decision. 

Author Response

The focus on those two aforementioned miRNAs is well explained from a theoretical perspective in the cited review [18] by the same group. However, I think it would be more appropriate if they have a data-driven explanation as to why they narrowed down to these two miRNAs compared to others. Therefore, I would like to see in the revised manuscript an expression analysis of all miRNAs expressed in the cohort and a correlation of each to different metrics of severity, as well as random metrics to validate the selection of those two miRNAs versus others.

A: We appreciate the extensive revision the reviewer has done and the comments: We added more information to the description of our miRNAs’ choice in the introduction section. On the other hand, following an exhaustive literature review, we select the miRNAs that could explain the variation of the COVID-19 phenotypes based on the miRNAs' expression and the genetic effects of the molecules. Unfortunately, we were not able to study additional miRNAs due to their high cost of analysis and financial difficulties during the pandemic. However, we consider that our study significantly contributes to the knowledge of the variability observed in the clinical outcome of COVID-19.

The survival conclusions should also be validated with the coxph statistical method. 

A: As suggested, we performed a Cox regression analysis, including age and miRNAs as covariates. The results were included on page 4, lines 104-108, page 14, line 414, and Supplemental Table 4, although a significant value was not observed.

The resolution of the figures should be improved (it looks like a copy-paste from R studio).

A: Thanks for the observation. We have just fixed the resolution and the color of the images.

Round 2

Reviewer 2 Report

The authors have addressed all my points. I believe the manuscript can be accepted for publication.

Author Response

Thank you for your comments and time in our manuscript.